# The sudden transition to online learning: Teachers' experiences of teaching during the COVID-19 pandemic

Elham Goudarzi[1]☯, Shirin Hasanvand [iD][2]☯*, Shahin Raoufi[2‡], Mitra Amini[3‡]

1 Student Research Committee, Lorestan University of Medical Sciences, Khorramabad, Iran, 2 Social Determinants of Health Research Center, School of Nursing and Midwifery, Lorestan University of Medical Sciences, Khorramabad, Iran, 3 Clinical Education Research Center, Shiraz University of Medical Sciences, Shiraz, Iran

☯ These authors contributed equally to this work.
‡ SR and MA also contributed equally to this work.
* hasanvand.sh1390@gmail.com

## Abstract

### Introduction

The sudden transition from face-to-face teaching to virtual remote education and the need to implement it during COVID-19 initially posed specific challenges to educational institutions. Identifying and understanding teachers' experiences pave the way for discovering and meeting educational needs. This study explored faculty members' teaching experiences during the COVID-19 pandemic.

### Materials and methods

The qualitative descriptive design via conventional content analysis was used. It was conducted from January 13, 2020, to May 10, 2022. In-depth interviews (online and in-person) of ten faculty members, three managers, and one staff from Lorestan University of Medical Sciences were conducted. They were purposefully selected with maximum variation. Simultaneously with data collection, analysis was performed using the approach Graneheim and Lundman proposed (2004). Lincoln and Goba's criteria were used to obtain the study's rigor.

### Results

Six categories emerged from the data: education in the shadow of the crisis, Challenges related to the teaching-learning process, Blurred boundaries between personal and professional lives, Positive consequences of e-learning, Trying to deal with the crisis, And dealing with the crisis.

### Conclusions

Initially, teachers faced several challenges in the teaching-learning process and even in their personal life. However, with time, the actions of the teachers and the managers caused

**Data Availability Statement:** All interview files are available from figshare, https://doi.org/10.6084/m9.figshare.23599155.v1.

**Funding:** The author(s) received no specific funding for this work.

**Competing interests:** The authors have declared that no competing interests exist.

**Abbreviations:** F2F, Face-to-face; ICT, Information and communications technology.

an increase in the quality of education. However, planning and foresight are needed in developing countries, including Iran, to appropriately face and optimally manage similar crises and move towards blended learning.

## Introduction

In late December 2019, a new subvariant of COVID-19 appeared in Wuhan, China, spreading rapidly worldwide [1]. The transmission rate of the virus and the subsequent pandemic were so significant that in the first month of 2020, the World Health Organization declared COVID-19 a state of emergency [2]. The first Iranian coronavirus case was officially announced in Qom on February 19, 2020 [3].

This global pandemic affected all areas of human life, including medical education [4], and disrupted face-to-face(F2F) learning worldwide. That is, F2F learning was suspended at medical universities, particularly in third-world countries, where the sudden change in educational planning was an inevitable consequence of the spread of the COVID-19 pandemic [5]. Insufficiency of resources and poor infrastructure severely damaged low-income countries [6]. In response to the closure of education, UNESCO recommended using distance education programs and related educational platforms by educational institutes to benefit from distance education and minimize any disruption in the learning-teaching process [7].

Distance education is a planned type of education where teaching and learning occur in different environments [8]. In recent years, the development of distance education has significantly contributed to promoting learning-teaching quality and expanding educational justice [9]. Certain advantages and disadvantages can be attributed to distance education. Its benefits include the feasibility of teaching regardless of time and place, cost-effectiveness [10], non-necessity of physical attendance [11], the existence of a variety of choices [12], saving time [10], studying simultaneously with working [12], and the development of participatory and independent types of learning [13]. Moreover, high probability of lack of concentration, the need for complex technology [12], reduced social interactions [14], unstable internet connection [13], the inability to comprehend and interpret students' facial expressions, and inability to hold practical and laboratory meetings are also among the disadvantages of distance education [6].

One important change after the campus closure was the transition from face-to-face universities to virtual universities to prevent the spread of COVID-19 [15]. In countries such as Italy, Spain, China, the USA, and even Brazil, distance learning has been widely used during the COVID-19 pandemic, particularly in medical education [16]. The use of distance education commenced in Iran with the emergence of the coronavirus. Although distance education was initially presented to students irregularly using social media, Iranian universities gradually adopted a systematic version of distance education using a centralized system of learning management according to the instructions issued by educational authorities. However, this system did not turn out to be an ideal one. Students and faculty members faced challenges such as slow internet speed, limited cyberspace to upload e-content, and reduced teacher-student interactions [17].

As was indicated in a study on the challenges encountered by faculty concerning distance education, a flawed organizational culture or the lack of a culture of working with e-learning tools such as computers, and the failure to train faculty members about how to operate these tools were also among the essential barriers to efficient distance education. Furthermore, lack of equipment, slow internet speed, lack of sufficient cyberspace for uploading educational

materials, being time-consuming and costly, and ignoring the intellectual property rights, such as unauthorized copying of the content and violation of copyright principles in the cyberlearning environment were other challenges raised by the participants [13]. In another study that focused on the experiences of teachers about distance education during the COVID-19 pandemic, many students expressed their concerns about losing contact with their peers, separation from academic communities, hardware and software inconveniences, lack of a quiet environment, and a separate room at home to attend virtual classes, lack of access to libraries and resources, and deprivation from clinical and laboratory activities. Faculty members complained about increased responsibility and workload and emphasized the necessity of having access to comprehensive mental health services that should be provided for both themselves and students [18]. In Marek et al. (2021) study, faculty who converted classes to remove learning during COVID-19 experienced much higher workloads and tension than in F2F classes [19].

A systematic review showed that fewer studies had been conducted in remote emergency teaching or e-learning during the Covid-19 pandemic [15]. Few studies, especially quantitative, have been conducted in Iran, and faculty members' experiences have yet to be investigated. This issue is a significant gap because it cannot be supposed that these experiences are similar to those in different cultural and social contexts. Thus, considering the emergence of the COVID-19 pandemic and its adverse effects on education worldwide, the disruption of teaching-learning processes in universities and teachers, and students' unpreparedness, it seems necessary to investigate teachers' experiences and deal with distance education. Such an investigation can contribute to the identification of the facilitators of and impediments to distance education if the COVID-19 pandemic persists or similar crises emerge. Thus, this study explored the teachers' teaching experiences during the COVID-19 pandemic.

## Materials and methods

### Design

This qualitative study used a conventional content analysis from January 13, 2020, to May 10, 2022, to describe the professor's teaching experiences during the COVID-19 pandemic. Content analysis is a study method for forming replicable and proper inferences from data to their context. It provides knowledge, new insights, a manifestation of facts, and a practical action guide. The aim is to acquire a condensed and comprehensive description of the phenomenon, and the result of the analysis is concepts or categories depicting the phenomenon. Conventional content analysis is used when there are no previous studies or research literature about the phenomenon, or it needs to be more cohesive. Researchers do not utilize predetermined categories. Rather, it lets categories and category labels flow from the data [20].

### Participants and data collection

Participants included ten faculty members, three administrators, and one staff of the Department of Education affiliated with Lorestan University of Medical Sciences (Western Iran). Participants were selected by purposeful and maximum variation sampling(gender, marital status, work experience, having experience in e-learning, specialty). Inclusion criteria were willingness to participate in the study and share their experiences.

The data were gathered through unstructured in-depth F2F or electronic interviews by the first author under the supervision of the second author. Since the second author was a faculty, the first author took responsibility for the interviews. However, the second author supervised the interviews because of his experience in qualitative research.

Due to the absence of some faculty members in the university, particularly at the beginning of the study, the interviews were conducted mainly electronically using either telephone or Adobe Connect video-conferencing software and later in person under health protocols. Also, F2F interviews were conducted with the participants' consent at their workplaces.

The interviews began with questions: "Could you please let us know about your teaching experiences since the emergence of the COVID-19 pandemic? What challenges did you face? How did you manage your class?" The participants were further investigated by answering probing questions such as "Could you explain more?" and "Could you please give an example?". The interviews were done individually and lasted an average of 20 to 50 minutes.

The data were recorded on a digital audio recorder. Field observations complemented the interviews. Sampling continued until data saturation when the collected data confirmed the previous data. Data saturation occurred after the 12th interview. To make sure two more interviews were conducted. Overall, 14 interviews were conducted with 14 participants. All the participants volunteered to participate in the study, and no one refused. The time and the place of the interviews (F2F) were arranged with the participants.

## Data analysis

The data analysis was conducted simultaneously by collecting data using the approach proposed by Granehim and Lundman(2004) with the following phases: 1) immediate transcription of interviews, 2) listening to them to obtain a general perception, 3) identification of significant parts and initial codes(the label of a meaning unit), 4) classification of similar initial codes in broader categories (creating categories), and 5) determination of the hidden content in the data [21]. Hence, after listening to the interviews, they were transcribed and read several times. In the next phase, the significant units were identified and coded. We consider a meaning unit as words, sentences, or paragraphs containing dimensions about each other through their content and context. The condensed meaning units were abstracted and labeled by a code. Subsequently, the codes were classified using a constant comparison technique, identifying differences and similarities, and subcategories were identified. Finally, the findings were compared, and categories were determined. Data analysis was carried out using the MAXQDA 10 software.

**Ethics approval and the consent to participate.** The present study was conducted under the Declaration of Helsinki. The code of ethics was also obtained from the Ethical Committee of the Vice-Chancellery for Research and Technology affiliated with Lorestan University of Medical Sciences (Code: I.R.LUMS.REC.1399.242). Providing the necessary explanations about the research objectives, we obtained written informed consent from all the participants. The subjects were allowed to record audio. The first author kept the recorded files in a locked file to ensure the security of the data.

## Rigor

Guba and Lincoln's (1994) criteria, i.e., credibility, dependability, confirmability, and transferability, were used to ensure the trustworthiness of the data. Participants with experience in the studied phenomenon were selected to increase the credibility of the data. The researcher's prolonged engagement and contact with participants (more than one year) were also considered. Moreover, more than one of the authors (the first and second authors) participated in the data analysis. Member-checking was also used. An attempt was made to improve data transferability by describing the participants' culture, context, and characteristics. The audit trail approach and maximum variation were used to ensure transferability. People with an experience in

qualitative research (Outside researchers) evaluated the data analysis process to ensure the findings were consistent.

The intercoder rater is a scale of the agreement between multiple coders about how similar data should be coded [22]. An inter-coder reliability analysis using Cohen's Kappa statistic was conducted to determine consistency between coders. Cohen's Kappa coefficient of agreement s was 0.871.

## Results

Fourteen participants participated in this study. Table 1 provides information about the participants.

The qualitative interviews extracted 1215 initial codes, 28 subcategories, and six main categories. Table 2 presents subcategories and categories extracted from the interviews.

### Category 1: Education in the shadow of the crisis

University closures necessitated the pursuit of distance education. Nevertheless, many faculty members and administrators thought they faced a temporary crisis. Thus, no significant measure was taken at the onset of the crisis. According to one of the participants, upon the emergence of the COVID-19 pandemic, it was maintained that the disease would disappear soon. Hence, no intense action was taken in the first month.

**1.1. The idea of being temporary.** Many teachers thought the crisis was temporary, so no particular action was taken early. One of the participants said:*"The disease would disappear soon, so no special action was taken in the first month of the disease"(P1).*

**1.2. Being unpredictable and uncertainty.** During this period, the teaching-learning process fluctuated, and faculty members were undecided. Moreover, some hospital wards were closed due to a lack of patients, decreased days of student internships, and a reduced variety of hospital cases. One of the participants said: *"In the first months of the crisis, we underwent extremely unpleasant experiences and did not know what to do in that situation.On the one hand, we were worried about the students.On the other hand, we had to reduce, for example, the*

**Table 1. Demographic and professional characteristics of the participants.**

| Participants | Gender | Age | Specialty | Rank | Professional Experience (years) | Marital Status | Interview Duration (min) |
|---|---|---|---|---|---|---|---|
| P1 | M* | 51 | Immunology | Associate professor | 18 | Married | 45 |
| P2 | F** | 42 | Nursing | Associate professor | 15 | Single | 50 |
| P3 | F | 55 | Nursing | Associate professor | 28 | Married | 37 |
| P4 | F | 44 | Neuroscience | Assistant professor | 26 | Married | 35 |
| P5 | F | 43 | Nursing | Assistant professor | 15 | Married | 20 |
| P6 | F | 45 | Higher Education Management | Assistant professor | 18 | Married | 42 |
| P7 | F | 45 | Obstetric surgeon | Assistant professor | 13 | Married | 20 |
| P8 | M | 44 | Nursing | Assistant professor | 20 | Single | 44 |
| P9 | M | 39 | Epidemiology | Assistant professor | 12 | Married | 32 |
| P10 | F | 47 | Anatomical sciences | Assistant professor | 7 | Married | 47 |
| P11 | F | 35 | Educational technology | Lecturer | 10 | Married | 34 |
| P12 | M | 49 | English language | Lecturer | 14 | Married | 35 |
| P13 | F | 40 | Reproductive health | Assistant professor | 18 | Single | 26 |
| P14 | F | 51 | Reproductive health | Associate professor | 26 | Married | 20 |

*Male

** Female

**Table 2. Categories and subcategories extracted from the interviews.**

| Categories | Subcategories |
| --- | --- |
| **1. Education in the Shadow of the Crisis** | The idea of being temporary |
| | Being unpredictable and uncertainty |
| | Educational confusion |
| | Life insecurity |
| | Teachers' concerns about the failure to learn from students |
| **2. Challenges related to the teaching-learning process** | Decreased quality of interpersonal interactions |
| | Authentication challenge |
| | Challenges related to online assessment |
| | Depreciation of teachers' equipment |
| | Teachers' low skills and knowledge of information and communications technology (ICT) |
| | Teachers' resistance to a sudden change in the teaching strategy |
| | Network connection issues |
| | Insufficient support |
| | The challenges in developing e-content |
| | Weaknesses in the implementation of practice-oriented training |
| | Students and teachers' Misuse of E-learning |
| **3. Blurred boundaries between personal and professional lives** | The long time required for distance education |
| | High workload |
| | Interference between work and nonwork roles and family restrictions |
| **4. Positive consequences of distance education during the COVID-19 pandemic** | Flexibility |
| | Facilitation of teaching-learning processes |
| | Cost reduction |
| **5. Try to deal with the crisis** | Using social media as a learning tool |
| | Targeted empowerment of faculty members |
| | Strategies for classroom management |
| | Strategies for assessing management |
| **6. Beyond dealing with the crisis** | Readiness to respond to future crises |
| | Search and discover solutions to strengthen the continuity of virtual education. |

number of internship days.Furthermore, there were almost no patients in the wards because peo-ple were full of fear and panic" (P2).

**1.3. Educational confusion.** The sudden transition in education from F2F to distance education confused teaching and reduced the quality of teaching. As one of the participants mentioned: *"Timely preparing all the educational content for students was another challenge for us.Inevitably, this preparation was prone to delay.I can assert that 99% of the faculty members could not provide the educational content for distance education without delay in the first semes-ter following the COVID-19 pandemic because all the courses were offered to students within the distance education framework" (p9).*

**1.4. Life insecurity.** Despite the widespread use of distance education in universities, some institutions face particular challenges. Medical students need to take internships in clini-cal settings. The fear of contracting the disease and the insecurity of life left many educators and administrators with moral and professional challenges. Some clinical faculty members

were dealing with the threats posed by COVID-19 in hospitals. Moreover, concerns about the spread of the disease adversely affected the quality and quantity of clinical sessions. As one of the faculty members said:

> "*in this period, the stress and tension were caused by the COVID-19 pandemic rather than teaching.I feel stressed out every time I attend class, but this time the stress was caused by the question of death or survival.*"
>
> *(p8)*

Another said:

> *"We were exposed to a death threat when we attended the hospital to teach students in this environment." Every day there were several confirmed COVID-19 cases in the hospital.I remember I contracted the coronavirus the day after examining a confirmed COVID-19 patient*
>
> *(P13).*

**1.5. Teachers' concerns about the failure to learn from students.** Some faculty members expressed concerns about students' failure to learn practical courses and the subsequent weakening of students' fundamental and practical knowledge and skills. One of the Midwifery faculty members stated: *"I was supposed to teach a 7th-semester student who has never been present in either a cesarean surgery or natural childbirth event in the hospital.This student will graduate in the following semester.I wonder if she can perform her duties as a hospital staff (p12).*

## Category 2: Challenges related to the teaching-learning process

There were several challenges regarding interpersonal interactions, assessment, network connection, and educational materials development in the teaching-learning process.

**2.1. Decreased quality of interpersonal interactions.** The lack of inclusive participation and direct interactions between teachers and students adversely affected the quality of education. A female faculty member said:

> *"Interactions, eye contact, and in any case, some emotional-psychological factors between teachers and students are eliminated in distance education"*
>
> *(P4).*

> *"The main problem we had with e-Learning was that we were not in touch with students, and they did not have the opportunity to visit us*"
>
> *(P8).*

**2.2. Authentication challenge.** Classroom management is complex in online classes due to the physical absence of teachers, particularly in large classes.

*"Coordination of the class time with students was one of the problems I encountered in distance education.Normally, students are expected to attend the class on time following our announcement of the class time.However, some students failed to attend the class on time"*

*(p5).*

*"It is difficult to coordinate all students in a virtual classroom" (p9).Another faculty member stated that "it is hard to control a class with a large population"*

*(p12).*

One of the significant concerns of faculty members was the problem of student identification and teachers' uncertainty about the presence of students in online classes. Some faculty members could not ensure the students' full-time presence in online classes: *"I do not know if the person taking the test is the student or someone else.Thus, identity verification is a major problem." (p1)*

**2.3. Assessment challenges.** One of the challenges for faculty members during this period was the limitation in preparing the test questions. They had to prepare new questions at the end of every semester due to the possibility of question leaks: *"Students had the chance to take screenshots of questions, which means that the questions could not be reused, and the faculty members had to redesign the questions at the end of every semester." (p1)*

Many faculty members considered students' copying of one anothers' homework, cheating, and negligence of ethical principles in preparing their assignments among the disadvantages of distance education: *"Concerning the disadvantages of distance education, I can safely assert that the most notable shortcoming of this type of education was the inaccurate assessment of students' knowledge.It is unclear who is taking the test.Is it the student, a friend, or someone who has been paid to take the test?" (p1).*

Obtaining unrealistic grades by cheating on each other and copying the answers to questions from the internet has led to a decline in the quality of education and the impossibility of distinguishing between intelligent and weak students. As one of the faculty members mentioned, *"A student with the overall average of grades that ranged from 14–15 has now the overall average of 19! Does it mean that all of them have become geniuses? What has happened? They are indeed cheating!" (p3).*

**2.4. Depreciation of teachers' equipment.** One of the requirements of distance education is to provide suitable hardware equipment. Some faculty members complained about the depreciation of equipment and personal computers due to frequent transportation to the university and their continuous use because of the lack of a proper hardware system to hold classes. *"The teachers had to use their equipment.Many of us did not have access to the necessary audio-visual equipment at the university" (p2).*

**2.5. Teachers' low skills and knowledge of ICT.** Due to the critical situation, teachers were forced to use distance education, while many were unfamiliar with virtual education technologies. This issue caused confusion and confusion in their minds. One of the professors said:

*"Professors have a problem with producing content, and how to upload it?" What exactly is this learning management system? Many professors do not know the system either"*

*(p3).*

One of *the* managers also confirmed this issue and said:

*"In the beginning, we had many problems because maybe 99% of our users were people who had not used the learning management system before this space and were not familiar with it."*

*(p11)*

**2.6. Teachers' resistance to a sudden change in the teaching strategy.** Some faculty members resisted this abrupt change in the teaching strategy: *"At the beginning of using Navid website (our native learning management system), initially our colleagues and then students resisted the use of this system" (p2).*
Another faculty member stated that;

*"The use of distance education had already commenced before the emergence of the COVID-19 pandemic, but most faculty members refused to teach within the framework prepared for distance education.They did not want to trouble themselves.Since they did not use the Navid LMS before the emergence of the crisis caused by the coronavirus, they needed help to cope with it during the crisis.They do not perform their tasks adequately.Every week, they were called to be reminded of their tasks*

*(p3).*

**2.7. Network connection issues.** The preparation of adequate infrastructure and the equipment required for communication and internet connection is highly significant in the development of distance education.

*"The lack of a high-speed internet connection was a significant problem adversely affecting this type of education.Another faculty member complained about the inconveniencies related to uploading the educational content, which was time-consuming that sometimes took several hours"*

*(p7).*

*"Moreover, low-speed internet at the university forced us to participate in online classes at home, where we had access to high-speed internet"*

*(p4).*

**2.8. Insufficient support.** Many faculty members complained about the lack of full-time support, a fundamental task of I.T. men. Hence, technical problems that were not resolved aggravated the situation: *"They did not respond to our hardware and software questions." (p6).*

**2.9. The challenges in developing e-content.** Faculty members were not skillful in ICDL. Thus, they were not familiar with content development technologies. The ultimate consequence was the production of non-standard content. Moreover, the lack of a quiet environment at home to record audio made it difficult for them to produce the content. One of the faculty members with previous experience in e-learning said: *"The teachers experience problems producing and uploading the content" (p3).*

A male faculty member who had twins said: "*I need a quiet room to produce the content at home, but I have twins who are almost one year old.The loud sound of their crying and playing was an obstacle to content production*"

(p12).

**2.10. Weaknesses in the implementation of practice-oriented training.**  Practical courses are highly significant in medical universities. However, due to the closure of universities, only theoretical courses were offered to students. Consequently, laboratory and clinical courses were not offered regularly. Furthermore, no standard simulators or special clinical and laboratory training tools existed. *"The difficulties with distance education aggravate fields of studies with many practical units, such as nursing, medical operating room technology, laboratory sciences, and medicine.Students' attendance in class is necessary for several of the courses of these fields, and the use of simulators cannot efficiently meet the requirements of practical courses" (P3).*

**2.11. Students and teachers' misuse of distance education.**  From the faculty members' perspective, students refused to attend online classes on time, participate attentively in the classes, study the electronic content on time, and do homework under various pretexts during the COVID-19 pandemic.

"Students prefer to have a classmate receive educational materials from their teachers to share them with them in the groups they create on social media networks such as WhatsApp and Telegram"

*(p2).*

*"Skyroom (a Persian version of Adobe Connect) could be an efficient software for distance teaching, but most students found different pretexts to justify their absence in online classes. They would claim that the teacher's voice is not clear, the video is frequently interrupted, our internet connection or the power has failed"*

*(p8).*

Unfortunately, there was no reliable monitoring system to evaluate faculty members' performance in the teaching-learning process, and this disadvantage caused several irregularities.

*"Some faculty members do not take e-learning seriously.The teacher, for instance, uploads a file while there is no content in the uploaded file"*

*(p3).*

*Some faculty members uploaded only three files for a two-credit course, while they were expected to prepare and upload at least 12 files*

*(p6).*

## 3. Blurred boundaries between personal and professional lives

According to the faculty, e-learning was time-consuming. It increased their workloads and interfered with their professional and personal roles. The faculty members said e-content development during the COVID-19 pandemic was time-consuming.

**3.1. Time-consuming distance education.**   content development during the COVID-19 pandemic was time-consuming and required time and energy, especially since the professors were not skilled enough. Some problems most professors mentioned were teacher involvement during non-office hours, time-consuming voicing of files, or re-voicing content due to hoarseness during voicing.*"If normally, I would check the slide a quarter of an hour before I go to class and leave, but not now! I would have to spend several hours now؛ Files and audio, that one and a half or two hours of my session now took four times that time" (p2)*

Another English teacher said, *"To prepare the file, I had a series of problems.I had to write the whole text of the book in English or scan it and convert it into a file in PowerPointFor each lesson, for example, I prepared 20 slides, each of which I had to write six to seven English lines; the interval when I was recording the sound was very difficult, meaning that I could, for example, mispronounce the word myself" (p12)*

One of the professors of epidemiology said: *"For example, I used to voice the complications of Quid on the files, and because the disease was unknown, I also talked about COVID, and then when we went forward, we saw that the risk factors for this disease had changed.The rate of change has changed, then the conversations have become old, and, for example, the prevention method has changed, and now we have to do this new-sounding file again.It was not that we had to leave the same file every semester; we had to update these files every time, which was time-consuming" (p9).*

**3.2. Increase workload.**   With the virtualization at once, the professors' workload increased significantly, development of standard content Increased the workload of the university's e-learning department; responding to faculty problems was one of the issues that multiplied the workload of faculty and staff.

One of the professors in charge of developing internship programs said:

*"It used to be that we had to put a program on the site from the beginning of the semester, but I can tell you that we wrote maybe 7 to 8 programs in the previous semester, and that put a lot of my work and that of my co-worker into the realm of bed and education".*

*(p2)*

*" As I said, we used to see students in class, but now we have to be online 24 hours and constantly answering to students, which caused us to devote much family time to this work "*

*(p6).*

**3.3. Interference between work and nonwork roles and family restrictions.**   The full-time presence of teachers at home and the introduction of virtual education led to role interference, resulting in changes in expectations and dissatisfaction among family members. Many professors confirmed this issue. This role interference was especially evident for teachers with younger children.*" Now our work problems have been brought to the family, both child and spouse.My child has been arguing with me many times in the virtual classroom because I said I am the head of the virtual classroom.He does not know his childish needs, But I have to be responsible because my work environment is one with home, which has diminished our mother's and my wife's roles" (p6).*

On the other hand, gathering family members at home due to corona restrictions caused noise and congestion in the home environment. They led to a lack of focus and increased psychological pressure on teachers to produce content and hold online classes. This issue led to

restrictions on family members, such as forcing children to remain silent. Consider deleting some family plans.

One of the teachers who had a young child said,

*"I will never forgive myself.I beat my children ،Because it happened in the middle of the sound; they suddenly entered the room and argued or, for example, asked a question "*

*(p7)*

*"When one's work environment becomes one with the living environment, many restrictions are created for the family.For example, I had to silence the whole family during class hours، The child should not watch the movie while the house was a space for rest, but because I had a virtual class or recorded content, for example, the noise of the environment was very annoying, and it made me put some restrictions on my family"*

*(p6)*

*"It was challenging, the family was distraught, that is, I had to tell the family to go to that room, and I would have the file in another room"*

*(p12).*

## 4. Positive consequences of e-learning

**4.1. Flexibility.** Despite the difficulties experienced and explained by the majority of the teachers, a few of them mentioned the benefits of e-learning. Flexibility is one of the features of e-learning. More precisely, the implementation of distance education does not depend on time and place. Moreover, it does not require a particular physical space shared by students and the teacher. Confirming the issue of flexibility in distance education, one of the faculty members said:

*"The most important advantage of distance education is that we can save time because many students who normally reside in dormitories no longer have to travel long distances to attend the campus.We can coordinate and hold classes at any time"*

*(p5)*

*"A remarkable advantage of e-learning is the chance peculiar to students employed in an organization.Many of our students who were employees during this period could benefit from e-learning and keep their jobs"*

*(p7).*

Moreover, although the faculty members encountered several challenges faced by distance education during the COVID-19 pandemic, this crisis and the consequent abrupt alteration in the education system forced the teachers who were negligent of e-learning to turn to this type of education, and to some extent performs their tasks willingly. One of the teachers confirmed the positive impact of the compulsory experience of distance education: *"Distance education had positive effects on us.That is, we would never be involved in distance education if the COVID-19 pandemic did not force us to carry out our tasks at home within the framework of e-learning.E-learning turned out to be highly beneficial.We learned how to use software" (p2).*

**4.2. Facilitation of educational processes.** *"If the COVID-19 crisis had not emerged, it might have taken ten years to reach this point, and we would have had to carry out a seven-*

*year mission in four to five months.Thus, in this respect, the COVID-19 pandemic might have been beneficial in this respect"*

*(p7).*

*"We had to learn how to use certain software, and I think it was a great chance for us".*

*(p4)*

**4.3. Cost reduction.**   Faculty members' experiences revealed that virtualization of education reduces the costs of holding workshops and conferences and provides students with accommodation and food. As one of the administrators mentioned in this regard: *"University expenses have significantly decreased.No longer was any fund allocated to routine services such as transportation, dormitory maintenance, and cooking, as well as serving food for students due to the closure following the aggravation of the crisis.Only a few students were still in dormitories to pursue their internship" (p5).*

## 5. Trying to deal with the crisis

**5.1. Using social media as a learning tool.**   Faculty members were unfamiliar with the university's learning management system at the onset of the coronavirus crisis. Hence, they used social media messaging applications to continue their teaching task and prevent interruption. One educational administrator stated, *"Due to the unpredictability of the continuation of the COVID-19 pandemic, it was first decided that the faculty members revise their PowerPoint files for a few sessions and send them to students via social media messaging applications or emails" (p1).*

**5.2. Targeted empowerment of faculty members.**   After familiarizing, teachers and students with the learning management system, several workshops and short-term courses were held to empower them. One of the teachers who were not competent in producing e-content said: *"Although the preparation of standard educational content by faculty members was an educational challenge, the difficulties and problems of content preparation decreased following the participation of the faculty members in the training workshops held by the Education Development Center (EDC) of the university" (p5).*

**5.3. Strategies for classroom management.**   Faculty members used various tricks to improve their class management following the alleviation of the COVID-19 crisis. They emphasized dividing the students in crowded classes into two groups, giving two distinct types of tests, setting periods for homework, giving appropriate and case-based questions, and using new teaching methods such as an online flipped classroom.

*"I tell the students that although the Navid LMS has a forum, Skyroom classes are live.Moreover, Skyroom provides us with a forum too.The cameras are connected within the framework of Skyroom, and we can see each other and share laptop screens"*

*(p10).*

*"I tried to design the questions so that they would not cheat.I would give them cases so that only those who study the sources pass the exam.I gave exams with conceptual questions"*

*(p14).*

**5.4. Strategies for assessment management.** To have a realistic and authentic assessment, some faculty members asserted that learners' assessment should not be limited to summative assessment. Furthermore, formative assessments and various assignments during the semester should be considered. *"I think student assessment should be carried out primarily during the semester and with the assignments they are given" (p1).*

## 6. Beyond dealing with the crisis

According to the faculty members and administrators, this crisis was beneficial in preparing higher education institutions for future crises. The COVID-19 pandemic foregrounded the significance of e-learning in educational institutions.

**6.1. Readiness to respond to future crises.** Although the COVID-19 pandemic imposed tremendous pressure on all aspects of society, including public health, specific potentials were gradually utilized to alleviate the coronavirus crisis. Consequently, the crisis was turned into appropriate educational opportunities. Several advantages can be attributed to the crisis management procedure during the COVID-19 pandemic. The pervasiveness of virtual education, the familiarity of teachers with various educational software, their capability to respond to future crises, and the development of specific strategies by the faculty members to guarantee the continuity of education by teachers are significant advantages. One of the teachers who were not familiar with educational software said: *"Virtual education was good in this era because it introduced us to different software anyway" (p2)."*

**6.2. Search and discover solutions to strengthen the continuity of virtual education.** Following the improvement in faculty members' capabilities in e-learning, their readiness to respond to future corona-like crises was also improved. If similar problems arise and e-learning becomes necessary, they will no longer have the concerns of previous crises. They can be more prepared to deal with and manage it.

*"The experience of e-learning during the COVID-19 pandemic contributed to our readiness to respond to similar crises in the future."*

*(p7).*

*"I started an e-learning course to prepare myself for similar crises because I thought it would not be our last experience in the coming years, and other similar issues are likely to occur in the future.If crises like coronavirus arise, we will hold classes virtually "*

*(P2).*

## Discussion

With the emergence of the COVID-19 pandemic, higher education suffered a great shock and, ultimately, an unexpected and unpredictable crisis worldwide, particularly in Iran. As a result, educational institutions were not ready to face this crisis, and the inevitable consequence was the emergence of education challenges.

In this study, the faculty members stated that they were initially unfamiliar with e-learning. This lack of familiarity and unpreparedness led to stress and mental confusion. Shenoy et al. (2020) and Aliyyah et al. (2020) reported similar instructors' experiences in their studies. The instructors who participated in these studies stated that they also were exposed to mental confusion at the beginning of the virtualization of education. However, they reported their relative satisfaction over time [23, 24].

On the other hand, the faculty members expressed concerns about the students' learning disabilities and the challenges related to the practical course. This issue was reported in the study by Salmani (2021) in a category entitled Students' superficial learning as a Challenge in E-learning in the COVID-19 Pandemic [25]. In another study, the low quality of virtual education compared to F2F teaching and students' superficial learning have been dealt with [26].

Other challenges from instructors' perspectives were raised consistent with previous studies. One of the primary challenges of e-learning based on the faculty members' experiences was their resistance to e-learning. Some faculty members did not believe in the efficacy of e-learning, which was consistent with the findings of the study conducted by Mohi et al. (2020) [27]. One of the reasons for faculty members' resistance to e-learning is that they are accustomed to traditional teaching. For years, they have preferred conventional teaching methods negligent of diversification or development of new methods of e-learning [28]. If the benefits of e-learning are emphasized, and faculty members become aware of this issue, as noted in the study by Mishra et al. (2021), they will be more motivated to teach [6].

The lack of high-speed internet bandwidth for preparing and uploading educational content led to faculty members' dissatisfaction with this type of teaching. Moreover, the lack of standard infrastructure and suitable hardware and software equipment were among the challenges encountered by other faculty members in this period. Using personal equipment led to the depreciation and multiple repairs of these devices. Rezaei et al. conducted a study reporting the challenges encountered by faculty members (2020). They stated that most teachers' cell phones were obsolete, and their laptops were primarily outdated [29]. The lack of suitable infrastructure was reported in many studies, particularly in developing countries [30, 31]. Teachers called for the support of experienced staff and experts concerning educational technologies and e-content production. Consistent with the present study, Dawn (2019) has pointed out the significance of teachers' familiarity with e-learning technologies during the COVID-19 pandemic [32]. Hence, the availability of a technical support team is essential for successfully developing an e-learning system and resolving the faculty members' problems. Investing in human resources and their training is a significant issue for the development of e-learning [31].

An essential part of the data indicated challenges related to the suspension practical and laboratory courses. In their study, Sam et al. (2020) also mentioned suspending clinical education activities to reduce disease transmission and decrease patient hospitalization [32]. This issue is fundamental. Many students may miss the opportunity to acquire communication and technical skills due to reduced internship periods, academic and laboratory activities, and the lack of standard tools for practical courses. Onyema et al. (2020) also cite the limited access to physical and laboratory facilities as an adverse effect of e-learning [26].

According to many teachers, the reduction in interpersonal interactions, the lack of F2F interactions, and the consequent delay in providing the students with feedback caused the faculty members and students not to know each other's abilities. In a similar study, the lack of proper communication between students and faculty members and their failure to provide their students with appropriate feedback was mentioned as teaching challenges during the COVID-19 pandemic [33]. According to De Oliveira et al. (2020), feedback difficulty in identifying students' strengths and weaknesses is sometimes due to the lack of modern communication between teacher and student [34].

Another study referred to the lack of efficient interactions in virtual education as the main negative feature of this type of education [35]. Thus, interactions between teachers and students and timely feedback are central to the success of the educational process.

Student identification was one of the challenges stated by faculty members to be reported in the present study. Similarly, Agarwal MS et al. (2020) reported that many faculty members

were uncertain about students' presence in the virtual classroom. The lack of accurate authentication tools was an obstacle to ensuring students' presence in virtual classrooms [36], which was consistent with the findings of this study. In the study by Lau et al. (2020), a defect in the evaluation of students by faculty members due to the lack of proper assessment tools was referred to as one of the most critical challenges of e-learning [37]. Other studies confirm that teachers can not guarantee that students are not cheating in electronic exams.

Moreover, it is impossible to ensure that the participant in the test is a student or someone else. Thus, the lack of proper assessment tools is one of the main problems of e-learning [28]. Virtualization of education also led to various kinds of abuse by some students and faculty members, consistent with the study by Agarwal et al. (2020) [36].

During this period, after overcoming the initial challenges of the COVID-19 crisis, the teachers' experience in the field of virtual education and how to manage virtual classes increased, which led to the use of a variety of strategies for optimal classroom management and electronic assessment, including variation in teaching methods, keeping the class enjoyable to students, student participation in discussions, randomization of questions, and giving conceptual and challenging questions. These strategies contribute to motivation enhancement in students to participate in virtual classes [24]. Moreover, being satisfied with a single assessment method reduces the validity of the assessment, and it is essential to use multiple assessment methods instead of summative assessment [35].

One of the most critical points related to teachers' experiences during this period was performing the teaching-learning task at home and blurring the boundaries between their personal and professional lives, which is consistent with Abedini et al. (2020) [38]. The home environment is not designed for academic activities. Hence, family members' involvement at home causes overlapping roles and dissatisfaction. A similar study indicated that with the full-time presence of teachers at home, family members' perceptions of the role of teachers changed. They expected teachers to perform the assigned functions alongside family members [39]. Teachers' workload was increased according to teachers' experiences, which was also confirmed in the study [40, 41].

Despite the many disadvantages and challenges of virtual education, particularly during the COVID-19 pandemic, some advantages can be attributed to this educational approach, such as independence about time and place, reduced costs, the possibility of studying and working simultaneously and saving time. These advantages were reported in the study by Mukherjee et al. (2021) [39]. Alterations in the attitude of teachers toward e-learning and adaptation to it was one of the main advantages and the main factor facilitating the educational process during the COVID-19 pandemic. Taghizadeh et al. asserted that the crisis was a blessing for faculty members. The teachers who were not interested in e-learning became familiar with various educational software [37].

Furthermore, the experience gained by faculty members has enabled them to deal with similar crises in the future. Vershitskaya et al. (2020) showed that following the onset of the COVID-19 pandemic, faculty members maintained that e-learning should not be ignored when the COVID-19 crisis is over. Instead, they asserted that it should be used as a supplement to F2F education [41].

Managers' attitudes were also considered in addition to the perspectives of faculty members in the present study, which could be considered a strength. However, the lack of students' opinions concerning e-learning was one of the limitations of this study. The researchers who conducted the present study are teachers. Thus, their assumptions might have biased their interpretations of the data. Consequently, we tried to validate the data analysis by recognizing and ignoring the irrelevant assumptions. Also, Mixed methods studies are suggested to understand e-learning problems in the post-covid era better.

These findings suggest that teachers must become trained the most proficient in e-learning and technology-enhanced learning, and this capability should continue. Special attention should be paid to open educational platforms. The use of e-learning as a response to covid has been accepted and continued. The place of learning in educational institutions should be reviewed. Support infrastructures for instructors and students should be provided. Managers and educational leaders should be encouraged and guided toward adaptive educational design.

## Conclusions

At the beginning of the outbreak of the Covid-19 pandemic, due to many reasons, including the resistance of teachers and learners to this approach and the lack of necessary infrastructure, the design and implementation of virtual education faced challenges. The teachers had many problems, such as weakness in class management, interference of roles, and even family restrictions. However, with time, the redoubled efforts of the teachers and the management of multiple roles by them, strengthening their knowledge in distance learning and e-content creation, and more participation in e-learning caused an increase in the quality of education. However, planning and foresight are needed in developing countries, including Iran, to appropriately face and optimally manage similar crises and move towards blended learning.

## Acknowledgments

This research was conducted within the framework of an MSc thesis. We want to thank all the participants.

## Author Contributions

**Conceptualization:** Elham Goudarzi, Shirin Hasanvand, Shahin Raoufi, Mitra Amini.

**Data curation:** Elham Goudarzi, Shirin Hasanvand.

**Formal analysis:** Elham Goudarzi, Shirin Hasanvand.

**Investigation:** Elham Goudarzi, Shirin Hasanvand.

**Methodology:** Elham Goudarzi, Shirin Hasanvand, Shahin Raoufi, Mitra Amini.

**Project administration:** Elham Goudarzi, Shirin Hasanvand.

**Supervision:** Shirin Hasanvand.

**Writing – original draft:** Elham Goudarzi, Shirin Hasanvand, Shahin Raoufi, Mitra Amini.

**Writing – review & editing:** Elham Goudarzi, Shirin Hasanvand, Shahin Raoufi, Mitra Amini.

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
