## [Decision Letter · Decision Letter 0]

24 Nov 2022

PONE-D-22-24937The Sudden Transition to Online Learning: Teachers' Experiences of Teaching during the COVID-19 PandemicPLOS ONE

Dear Dr. Hasanvand,

Thank you for submitting your manuscript to PLOS ONE. After careful consideration, we feel that it has merit but does not fully meet PLOS ONE’s publication criteria as it currently stands. Therefore, we invite you to submit a revised version of the manuscript that addresses the points raised during the review process.

This manuscript is well-written. Improvement in methods section and conclusion section is suggested. Conclusions must be alligned with results & discussion. You can find reviewers report and make necessary revision in the manuscript.

We look forward to receiving your revised manuscript.

Kind regards,

Apurva Kumar Pandya, PhD

Academic Editor

PLOS ONE

Journal Requirements:

Additional Editor Comments:

This manuscript is well-written. Improvement in methods section is suggested:

1- Theme network resulting from three-stage coding

2- Table of agreement between two coders or calculation of kappa coefficient

3- The interview questions should come in the tools and methods section

Reviewers' comments:

Reviewer's Responses to Questions

**Comments to the Author**

1. Is the manuscript technically sound, and do the data support the conclusions?

Reviewer #1: Yes

2. Has the statistical analysis been performed appropriately and rigorously? 

Reviewer #1: Yes

3. Have the authors made all data underlying the findings in their manuscript fully available?

Reviewer #1: No

4. Is the manuscript presented in an intelligible fashion and written in standard English?

Reviewer #1: Yes

5. Review Comments to the Author

Reviewer #1: This manuscript is well-written. To improve the methodological part, the following are suggested:

1- Theme network resulting from three-stage coding

2- Table of agreement between two coders or calculation of kappa coefficient

3- The interview questions should come in the tools and methods section

6. PLOS authors have the option to publish the peer review history of their article (what does this mean?). If published, this will include your full peer review and any attached files.

Reviewer #1: No

---

## [Author Response · Author response to Decision Letter 0]

8 Jan 2023

We thank the reviewer for his/her valuable comments. These comments will improve our manuscript. We answered the comments point by point. Thank the editor for following up and speeding up the work.

---

## [Decision Letter · Decision Letter 1]

26 Apr 2023

PONE-D-22-24937R1The Sudden Transition to Online Learning: Teachers' Experiences of Teaching during the COVID-19 PandemicPLOS ONE

Dear Dr. Hasanvand,

Thank you for submitting your manuscript to PLOS ONE. After careful consideration, we feel that it has merit but does not fully meet PLOS ONE’s publication criteria as it currently stands. Therefore, we invite you to submit a revised version of the manuscript that addresses the points raised during the review process. Please submit your revised manuscript by Jun 10 2023 11:59PM. If you will need more time than this to complete your revisions, please reply to this message or contact the journal office at plosone@plos.org. Please include the following items when submitting your revised manuscript:A rebuttal letter that responds to each point raised by the academic editor and reviewer(s). You should upload this letter as a separate file labeled 'Response to Reviewers'.A marked-up copy of your manuscript that highlights changes made to the original version. You should upload this as a separate file labeled 'Revised Manuscript with Track Changes'.An unmarked version of your revised paper without tracked changes. You should upload this as a separate file labeled 'Manuscript'.If applicable, we recommend that you deposit your laboratory protocols in protocols.io to enhance the reproducibility of your results. Protocols.io assigns your protocol its own identifier (DOI) so that it can be cited independently in the future. For instructions see: https://journals.plos.org/plosone/s/submission-guidelines#loc-laboratory-protocols. Additionally, PLOS ONE offers an option for publishing peer-reviewed Lab Protocol articles, which describe protocols hosted on protocols.io. Read more information on sharing protocols at https://plos.org/protocols?utm_medium=editorial-email&utm_source=authorletters&utm_campaign=protocols.

We look forward to receiving your revised manuscript.

Kind regards,

Apurva Kumar Pandya, PhD

Academic Editor

PLOS ONE

Journal Requirements:

Additional Editor Comments:

Thanks for submitting revised manuscript. You have addressed major points; however, it requires further revision. We have received reviewers comments and I agree with their feedback. Authors need to address reviewers comments and submit.

Reviewers' comments:

Reviewer's Responses to Questions

**Comments to the Author**

1. If the authors have adequately addressed your comments raised in a previous round of review and you feel that this manuscript is now acceptable for publication, you may indicate that here to bypass the “Comments to the Author” section, enter your conflict of interest statement in the “Confidential to Editor” section, and submit your "Accept" recommendation.

Reviewer #1: (No Response)

Reviewer #2: (No Response)

2. Is the manuscript technically sound, and do the data support the conclusions?

Reviewer #1: Yes

Reviewer #2: Partly

3. Has the statistical analysis been performed appropriately and rigorously? 

Reviewer #1: N/A

Reviewer #2: No

4. Have the authors made all data underlying the findings in their manuscript fully available?

Reviewer #1: Yes

Reviewer #2: No

5. Is the manuscript presented in an intelligible fashion and written in standard English?

Reviewer #1: Yes

Reviewer #2: Yes

6. Review Comments to the Author

Reviewer #1: Dear author,

Thank you for your effort and hope you succeed. I think this paper is valuable but the methodology is a very critical part of each scientific paper. I feel this section hasn't been written properly, although it is clear that you have done your study scientifically. But readers are the priority and it is important when they read your paper what they learn and understand.

Studying interviews and identifying people's opinions in a scientific article requires correct methodology. It is clear that interviews were conducted, but these interviews were reviewed without structure. If there is a theme analysis method, it is not clear what approach it is. These things you showed in the findings by mentioning the interview are more appropriate for a scientific worksheet, not methodology. If phenomenology was used, we would get a better understanding of your methodology. Also, the synthesis research method, which is the comparison of interviews with valid theories, could form a scientific research paper. But the main focus of the article, which is its methodology, does not have a proper process and structure in this research. for example, if you have done this part according to the grounded theory, it is needed to mention the reliability, and the steps of coding. Also, you stated your population was 14 people. Wasn't there any saturation? or was the information you obtained enough? wasn't there a need to spread your scope of interviews?

You may need to add 1 or 2 paragraphs to describe the interview coding step by step.

Reviewer #2: The Sudden Transition to Online Learning: Teachers' Experiences of Teaching during the COVID-19 Pandemic

Overall, the paper is not well written.

In the abstract the method employed is missing even as a perspective paper the methodology needs to be well specified

The introduction section needs to be improved to include the main research problems and research questions.

The introduction section can be improved. The main issues and background of the study needs to be well discussed as related to the literature.

Also, improve the research questions to be explored in the introduction section.

A section on literature review or related works needs to be written. Include prior studies on Teachers' Experiences of Teaching during the COVID-19 Pandemic. Studies as shown below can be included;

Anthony Jnr, B., & Noel, S. (2021). Examining the adoption of emergency remote teaching and virtual learning during and after COVID-19 pandemic. International Journal of Educational Management, 35(6), 1136-1150.

Marek, M. W., Chew, C. S., & Wu, W. C. V. (2021). Teacher experiences in converting classes to distance learning in the COVID-19 pandemic. International Journal of Distance Education Technologies (IJDET), 19(1), 89-109.

Anthony Jnr, B. (2022). An exploratory study on academic staff perception towards blended learning in higher education. Education and Information Technologies, 27(3), 3107-3133.

Mercier, K., Centeio, E., Garn, A., Erwin, H., Marttinen, R., & Foley, J. (2021). Physical education teachers’ experiences with remote instruction during the initial phase of the COVID-19 pandemic. Journal of Teaching in Physical education, 40(2), 337-342.

Besides, I feel the literature review is weak

Moreover, how does your current study differ from the studies provided above

The discussion section needs to map back to the literature.

What are the practical and research implications from the study

Expand more on the limitations and future works.

7. PLOS authors have the option to publish the peer review history of their article (what does this mean?). If published, this will include your full peer review and any attached files.

Reviewer #1: No

Reviewer #2: No

While revising your submission, please upload your figure files to the Preflight Analysis and Conversion Engine (PACE) digital diagnostic tool, https://pacev2.apexcovantage.com/. PACE helps ensure that figures meet PLOS requirements. To use PACE, you must first register as a user. Registration is free. Then, login and navigate to the UPLOAD tab, where you will find detailed instructions on how to use the tool. If you encounter any issues or have any questions when using PACE, please email PLOS at figures@plos.org. Please note that Supporting Information files do not need this step.<quillbot-extension-portal></quillbot-extension-portal>

---

## [Author Response · Author response to Decision Letter 1]

28 Apr 2023

Additional Editor Comments:

Thanks for submitting the revised manuscript. You have addressed major points; however, it requires further revision. We have received the reviewer's comments, and I agree with their feedback. Authors need to address the reviewer's comments and submit.

Reviewers' comments:

Reviewer's Responses to Questions

Comments to the Author

We thank the reviewers for their valuable comments. These comments will improve our manuscript.

1. If the authors have adequately addressed your comments raised in a previous round of review and you feel that this manuscript is now acceptable for publication, you may indicate that here to bypass the “Comments to the Author” section, enter your conflict of interest statement in the “Confidential to Editor” section, and submit your "Accept" recommendation.

Reviewer #1: (No Response): -

Reviewer #2: (No Response): -

2. Is the manuscript technically sound, and do the data support the conclusions?

Reviewer #1: Yes

Reviewer #2: Partly

3. Has the statistical analysis been performed appropriately and rigorously?

Reviewer #1: N/A

Reviewer #2: No

In the statistical analysis section, we talked about the data analysis method in detail. At first, the data analysis method and its steps are described. In the following, we have said what we have done at each step. We added some explanations again. I hope it is clear and clears our confusion. The method of data analysis is well-known in nursing and is frequently used by researchers in qualitative content analysis. The reference is also introduced. We used Graneheim and Lundman (2004) method. They have said our paper is intended to be used in nursing research and education and to contribute to a debate on qualitative content analysis.

4. Have the authors made all data underlying the findings in their manuscript fully available?

The PLOS Data policy requires authors to make all data underlying the findings described in their manuscript fully available without restriction, with rare exceptions (please refer to the Data Availability Statement in the manuscript PDF file). The data should be provided as part of the manuscript or its supporting information or deposited in a public repository. For example, in addition to summary statistics, the data points behind means, medians, and variance measures should be available. If there are restrictions on publicly sharing data—e.g., participant privacy or use of data from a third party—those must be specified.

Reviewer #1: Yes

Reviewer #2: No

Response: All study data (interviews with participants) are available in the manuscript and can be accessed.

As you can see in the above sentence, in the answer to the previous review, the possibility of accessing the data is clearly mentioned.

5. Is the manuscript presented in an intelligible fashion and written in standard English?

Reviewer #1: Yes

Reviewer #2: Yes

6. Review Comments to the Author

Reviewer #1: Dear author,

Thank you for your effort, and I hope you succeed. I think this paper is valuable but the methodology is a critical part of each scientific paper. I feel this section hasn't been written properly, although it is clear that you have done your study scientifically. But readers are the priority, and it is important when they read your paper what they learn and understand. Studying interviews and identifying people's opinions in a scientific article requires correct methodology. It is clear that interviews were conducted, but these interviews were reviewed without structure. If there is a theme analysis method, it is not clear what approach it is.

Very thanks for your valuable comment. We hope that our response is clear. This study uses content analysis as a qualitative descriptive approach. Thematic analysis has not been used. Although the boundaries between these two approaches are unclear and often used interchangeably, there still needs to be more clarity about their similarities and differences. According to this comment, the research team decided to fully describe the study design. Thank you for your detailed comment. It was corrected. Also, according to the reviewer’s opinion, the type of interviews was changed from semi-structured interview to unstructured one.

 These things you showed in the findings by mentioning the interview are more appropriate for a scientific worksheet, not methodology. If phenomenology was used, we would get a better understanding of your methodology. Also, the synthesis research method, which is the comparison of interviews with valid theories, could form a scientific research paper. But the main focus of the article, which is its methodology, does not have a proper process and structure in this research. for example, if you have done this part according to the grounded theory, it is needed to mention the reliability and the steps of coding. 

In qualitative research, several analysis methods can be used, for example, phenomenology, hermeneutics, grounded theory, ethnography, phenomenography, and content analysis. In contrast to qualitative research methods, qualitative content analysis is not linked to any particular science, and there are fewer rules to follow. Therefore, the risk of confusion in philosophical concepts and discussions is reduced. No matter the chosen method, the analysis process reduces the volume of text collected, identifies and groups categories together, and seeks some understanding of it. In some way, the researcher attempts to “stay true” to the text and to achieve trustworthiness. In this study, the subject of trustworthiness has been given special attention by using the content analysis approach. (Bengtsson M. How to plan and perform a qualitative study using content analysis. Nursing Plus Open. Volume 2, 2016, 8-9)

We thought the methodology was not well explained and confusing for the readers. Therefore, the required corrections were applied in the methodology section. In other word, because the study design was not well described, the findings were not concrete. We hope that by providing explanations about the study method and the purpose of content analysis, the ambiguity has been resolved.

Also, you stated your population was 14 people. Wasn't there any saturation? or was the information you obtained enough? wasn't there a need to spread your scope of interviews?

It was corrected, and the necessary explanations were added in the methodology section.

You may need to add 1 or 2 paragraphs to describe the interview coding step by step.

Previously, the data analysis method and its steps have been mentioned in the data analysis section. We have mentioned each step exactly. Again, other explanations were added.

Reviewer #2: The Sudden Transition to Online Learning: Teachers' Experiences of Teaching during the COVID-19 Pandemic

Overall, the paper is not well written.

In the abstract, the method employed is missing. Even as a perspective paper, the methodology needs to be well specified.

Thanks for your valuable comment. It was mentioned in paragraph 6.

The introduction section needs to be improved to include the main research problems and research questions. The introduction section can be improved. The main issues and background of the study needs to be well discussed as related to the literature. Also, improve the research questions to be explored in the introduction section. A section on literature review or related works needs to be written. Include prior studies on Teachers' Experiences of Teaching during the COVID-19 Pandemic. Studies as shown below can be included;

Anthony Jnr, B., & Noel, S. (2021). Examining the adoption of emergency remote teaching and virtual learning during and after COVID-19 pandemic. International Journal of Educational Management, 35(6), 1136-1150.

Marek, M. W., Chew, C. S., & Wu, W. C. V. (2021). Teacher experiences in converting classes to distance learning in the COVID-19 pandemic. International Journal of Distance Education Technologies (IJDET), 19(1), 89-109.

Anthony Jnr, B. (2022). An exploratory study on academic staff perception towards blended learning in higher education. Education and Information Technologies, 27(3), 3107-3133.

Mercier, K., Centeio, E., Garn, A., Erwin, H., Marttinen, R., & Foley, J. (2021). Physical education teachers’ experiences with remote instruction during the initial phase of the CoVID-19 pandemic. Journal of Teaching in Physical Education, 40(2), 337-342.

Besides, I feel the literature review is weak. Moreover, how does your current study differ from the studies provided above. The discussion section needs to map back to the literature.

Thanks for the above papers. We use them in the introduction for the completion of the literature review. According to the review’s opinion, changes were made to the problem statement. In the following, planned distance learning is discussed first. Then, emergency distance learning with the arrival of Corona and its challenges, especially for professors, are mentioned. The respected review’s references were used for the literature review. The contents of the discussion section have been prepared and highlighted regarding 18 new references (17 references from 2020 to 2021). The discussion related to each category has been done separately by comparing the results of this study with other studies related to Covid.

What are the practical and research implications of the study?

It was added.

Expand more on the limitations and future works.

It was added.

7. PLOS authors have the option to publish the peer-review history of their article (what does this mean?). If published, this will include your full peer review and any attached files.

If you choose "no," your identity will remain anonymous, but your review may still be made public.

Do you want your identity to be public for this peer review? For information about this choice, including consent withdrawal, please see our Privacy Policy.

Reviewer #1: No

Reviewer #2: No

---

## [Editor Report · Decision Letter 2]

7 Jun 2023

The Sudden Transition to Online Learning: Teachers' Experiences of Teaching during the COVID-19 Pandemic

PONE-D-22-24937R2

Dear Dr. Hasanvand,

We’re pleased to inform you that your manuscript has been judged scientifically suitable for publication and will be formally accepted for publication once it meets all outstanding technical requirements.

Kind regards,

Muhammad Arsyad Subu, Ph.D

Academic Editor

PLOS ONE

Additional Editor Comments (optional):

Reviewers' comments:

<quillbot-extension-portal></quillbot-extension-portal>

---

## [Editor Report · Acceptance letter]

11 Jul 2023

PONE-D-22-24937R2 

The Sudden Transition to Online Learning: Teachers' Experiences of Teaching during the COVID-19  Pandemic 

Dear Dr. Hasanvand:

I'm pleased to inform you that your manuscript has been deemed suitable for publication in PLOS ONE. Congratulations! Your manuscript is now with our production department. 

Kind regards, 

on behalf of

Dr. Muhammad Arsyad Subu 

Academic Editor

PLOS ONE